# Complex relationship between crop yields and crop growing period: The shortened growing period before flowering contributes to yield increase in common buckwheat (*Fagopyrum esculentum*)

**Gen Sakurai**[ID]*, **Naoki Ishitsuka, Norikazu Okabe**

Institute for Agro-Environmental Sciences, National Agriculture and Food Research Organization, Tsukuba, Ibaraki, Japan

* sakurai.gen270@naro.go.jp

## Abstract

Predictions from process-based crop models have suggested that shorter growing seasons due to increases in temperature will lead to reductions in crop yields. However, a study to assess this relationship using statistical data would not be sufficient. In this study, a statistical analysis was carried out using historical crop calendar data and yield data for common buckwheat (*Fagopyrum esculentum*) to investigate how increased temperature affects crop yields through changes in the growing season. First, the parameters of the model representing the relationship between weather and growth rate were estimated using crop calendar data in Japan. Second, the relationship between climate factors and yield was estimated using the generalized additive model. We then examined how rising temperatures under future weather conditions would affect yield through changes in buckwheat growth rate. The results suggested that integrated solar radiation before flowering had a negative effect on buckwheat yield, while integrated solar radiation after flowering had a positive effect on yield. It was suggested that the growth rate of buckwheat was faster at higher temperatures and slower at longer day lengths. Under future climate conditions, higher temperatures and shorter pre-flowering periods were predicted to result in longer post-flowering day lengths and more integrated post-flowering solar radiation due to a longer post-flowering growing season. As a result, the increase in growth rate due to increased temperatures had a positive effect on yield outweighed the slight negative effect of the temperature increase after flowering. Based on historical statistical data, this study analyzed the complex effects of phenology changes due to increased temperature on crop yields, and similar analyses are expected to be conducted for other crops in the future.

**Data availability statement:** The code used to perform the analyses presented in this paper

is publicly available on GitHub at ([https://github.com/gen-git/BuckWheatAnalysis/releases/tag/v1.0](https://github.com/gen-git/BuckWheatAnalysis/releases/tag/v1.0)). This repository includes all scripts necessary to reproduce the results and figures. Due to their large size, the meteorological and yield data are not included in the GitHub repository. These datasets are archived separately on Zenodo at ([https://zenodo.org/records/15868967](https://zenodo.org/records/15868967)). If you require further assistance, please contact the authors.

**Funding:** This research was funded by the Environment Research and Technology Development Funds (JPMEERF20S11803, JPMEERF25S12421 and JPMEERF20222G01) of the Environmental Restoration and Conservation Agency, which is under the Ministry of the Environment of Japan. This work was also supported by JSPS KAKENHI Grant Numbers 24H00571 and 21H02294.

**Competing interests:** The authors have declared that no competing interests exist.

## Introduction

Climate change due to global warming has been already observed worldwide since the mid-20th century, and its existence is no longer in doubt. [1]. Among the sectors that can be affected by climate change, crop production is directly affected by climatic conditions such as temperature, solar radiation, and precipitation. The increase or decrease in crop yields due to climate change is one of the main issues in climate change impact studies. In particular, for staple crops such as wheat, rice, maize, and soybean, which account for the majority of global crop consumption, there have been a large number of studies on the relationship between climatic factors and crop productivity [2,3]. In addition to these major crops, a wide variety of other crops are produced worldwide and may be affected by climate change. Predictions of the effects of climate change on the yields of a wide range of crops, including minor crops, provide useful information for farmers, consumers, and policy makers.

Numerous studies have been conducted to predict the potential change in crop yields due to climate change. In these studies, process-based crop models have been the primary tool for assessing the impact of climate change on crop yields [4–15], in which each of the crop growth processes, such as photosynthesis and evapotranspiration, is calculated according to daily climatic factors. Although the increase in temperature due to climate change can affect crop productivity through different pathways, the increase in temperature generally leads to an increase in growth rate. In many cases, an increase in growth rate has a negative effect on crop yield because an increase in growth rate shortens the growing period and reduces the total amount of solar radiation intercepted by crops during the growing period [16–22].

However, shorter growing periods may not necessarily lead to lower crop yields. For example, the reproductive period after flowering is one of the most sensitive periods for many crops [23–25], and shifting the flowering period by shortening the vegetative period may allow crops to avoid low temperatures after the summer season. This is just one possible example of how changing growing periods can potentially increase crop yields. There are likely to be complex relationships between growing periods and crop yields. Understanding the relationship between growing periods and crop yields is crucial to assessing the impact of climate change on crop productivity. For this purpose, it should be useful to analyze the relationship using actual statistical data on crop yields and crop calendars.

In Japan, since the 1960s, the Ministry of Agriculture, Forestry and Fisheries (MAFF) has been collecting extensive statistical data on crop yields and areas at municipal, prefectural, and national levels. These data have been recorded for various crops, including cereals, vegetables, fruit trees, grasses, and flowers. In addition to yields and areas, data on the crop calendar are also available. These data allow us to assess the effect of change in the growing periods on crop yields using statistical methods for even minor crop species.

The purpose of this study is to evaluate the effect of changes in the growing period due to global warming on common buckwheat (*Fagopyrum esculentum*) yield using statistical crop data. Among the minor crop species, common buckwheat is planted almost all over Japan, and the statistical information on the crop calendar is also available. According to Food and Agriculture Organization of the United Nations [26], about 1.88 million tons of buckwheat are produced worldwide, and more than 50% of those are produced in cold regions such as Russia and Ukraine. As global warming proceeds, warmer regions will expand. Therefore, assessing the possible response due to the change in the growing period is one of the important issues in the context of climate change.

In this study, we first construct the model that estimates the length of the growing periods, in which temperature and day length are considered using historical crop calendar data. Second, we analyze the relationship between buckwheat yields and climate factors during each

growing period (before or after flowering) using historical yield data. Third, by estimating the growing periods and crop yield under future climate conditions, we evaluate the effect of climate change on buckwheat yield and discuss the effect of the change of the growing periods on the yields under future climate conditions. The recent IPCC report [3] highlights the lack of studies on the relationship between climate factors and the yield of minor crops. This study would also help to add one of the minor crops to the list of crops for which the future impacts of climate change are being assessed.

## Materials and methods

### Past crop yield and calendar data

The data on yield and harvested area of common buckwheat (*Fagopyrum esculentum*) from 1993 to 2020 and the crop calendar for 1995 were obtained from the statistics on crops of the Japanese Ministry of Agriculture, Forestry and Fisheries (https://www.maff.go.jp). The spatial resolution of the crop yield and harvested area data are recorded at the municipal level (in terms of administrative districts in Japan, prefectures include municipalities and counties, while municipalities include cities, towns, and villages). The spatial resolution of the crop calendar data is prefecture-level for four growing events: sowing, germination, flowering, and harvesting. Out of 47 prefectures in Japan, these data do not include data from Osaka, Wakayama, and Okinawa prefectures, where buckwheat data have not been recorded, so these three prefectures were excluded from the analysis. The total number of crop yield data was 26776 (the average yield of the data is 65.51 kg per 10 ares). The crop yield data differed over the years because the numbers of municipalities planting buckwheat differed. The minimum sample size of crop yield data was 650 in 1994 (the average is 92.04 kg per 10 ares), and the maximum was 2013 in 2001 (the average is 69.80 kg per 10 ares).

### Past climate data

Meteorological data were obtained from the Agro-Meteorological Grid Square dataset of the National Agriculture and Food Research Organization [27] (NARO, https://amu.rd.naro.go.jp). This data set provides 1 km mesh meteorological data, which is constructed from observation data of the Japan Meteorological Agency. In this dataset, the observation data was interpolated by using the information on elevation and other factors [27] for making 1 km mesh data. This study used data on daily precipitation, average temperature, and total solar radiation. Because the spatial resolution of the statistical yield data is at the municipality level, we calculated the average meteorological values for each day for each municipality, considering the area of farmland. Municipality-level administrative district data and land use information data for all of Japan were obtained from the National Land Information Download Service by the Japanese Ministry of Land, Infrastructure and Transport [28] (accessed July 6, 2021) (https://nlftp.mlit.go.jp/ksj/). Farmland area for buckwheat was defined as the sum of the area of rice fields and the area of other agricultural land in the data set because buckwheat is also cultivated in the rice fields as the conversion crop of rice in Japan.

### Future climate data

Predicted meteorological data for the periods 2020–2022 and 2051-2060 were obtained from [29] (ver.1, accessed October 19, 2021). This dataset provides bias-corrected meteorological data, including precipitation, solar radiation, and temperature, of five Global Climate Models (GCMs) (MIROC6, MRI-ESM2-0, ACCESS-CM2, IPSL-CM6A-LR, and MPI-ESM1-2-HR) as daily data with approximately 1 km spatial resolution for Japan region. We used the data

on daily precipitation, average temperature, and total solar radiation predicted from the five GCMs under the SSP5-RCP8.5 scenario. We adopted only this warmest scenario because this study aims to untangle the relationship between growing periods and crop yields, not to estimate future buckwheat yields and those uncertainties. The average meteorological values were calculated for each day and each municipality, considering the farmland area as with the past climate data. The predicted future land use scenario was obtained from [30] for calculating the average values. Farmland area was defined as the sum of the area of rice fields and the area of other agricultural land in the data set as with the past climate data.

## Model for estimating growing periods

To predict the growing periods for buckwheat, the parameters of the following equation, which calculates the growth rate based on temperature and day length [31], were estimated using the crop calendar data and the past climate data.

$$
\begin{aligned}
R_{i,t,d} &= \frac{1 - \exp\left(B\left(L_{i,t,d} - L_c\right)\right)}{\left(1 + \exp\left(A\left(T_{i,t,d} - T_n\right)\right)/G\right)}, \quad \left(when \quad L_{i,t,d} < L_c\right) \\
R_{i,t,d} &= 0, \quad \left(when \quad L_{i,t,d} \geq L_c\right)
\end{aligned}
\tag{1}
$$

where $R_{i,t,d}$ is the growth rate at municipality $i$, year $t$, and day $d$. $L_{i,t,d}$ is day length, $T_{i,t,d}$ is the mean temperature, $A$, $B$, $L_c$, $T_n$, and $G$ are parameters to be estimated. Though this model was originally constructed for rice [31], it is known that this shape of function well captures the relationship between temperature, day length, and the growth rate. If the summation of the growth rates ($\sum_d R_{i,t,d}$) reaches a threshold $\phi$ ($\phi$ = 1 in this study), the crop moves to the next stage (flowering or maturity). In this study, the timing of maturity is assumed to be the timing of harvest. Because we separated the period into two growing periods (before flowering and after flowering), we estimated the parameters for each period, respectively. The day length of each day was estimated approximately from latitude and solar declination [32]. For estimating the parameters, we used "optim" function of R version [33]. To obtain stable parameter values, we conducted a bootstrapping method in which the data on sowing, flowering, and harvesting of the municipalities were randomly sampled, and then the parameter values that minimize the sum of the differences between estimated flowering days (or harvesting days) and observed ones (the root mean square errors between the observation and estimation). This procedure was repeated 300 times, and then we used the average values of the obtained parameters for the prediction of future crop yields.

## Statistical analysis

We constructed the generalized additive model to analyze the relationship between the yield of buckwheat and the climatic factors using the historical yield data from 1993 to 2020. A Generalized Additive Model (GAM) is a flexible regression model that allows the relationship between explanatory and response variables to be represented as smooth, nonlinear functions. It extends generalized linear models by replacing linear terms with smooth functions, improving the model's ability to capture complex patterns in the data [34]. The explained variable was the yield of buckwheat, and the explanatory variables were temperature, precipitation, solar radiation, year, and harvested area in each municipality. All explanatory variables were assigned as fixed effects. When incorporating climatic values, the period from sowing to harvest was divided into two growing periods: the period before flowering (sowing to flowering) and the period after flowering (flowering to harvesting), as with the model of the growth rate.

The generalized additive model that includes all explanatory variables described below:

$$\begin{aligned}
y_{i,t} &= f_{\mathrm{T,BF}}(T_{\mathrm{BF},i,t}) + f_{\mathrm{P,BF}}(P_{\mathrm{BF},i,t}) + f_{\mathrm{S,BF}}(S_{\mathrm{BF},i,t}) \\
&\quad + f_{\mathrm{T,AF}}(T_{\mathrm{AF},i,t}) + f_{\mathrm{P,AF}}(P_{\mathrm{AF},i,t}) + f_{\mathrm{S,AF}}(S_{\mathrm{AF},i,t}) \\
&\quad + f_{\mathrm{F}}(F_{i,t}) + f_{\mathrm{t}}(t) + \epsilon_{i,t},
\end{aligned} \tag{2}$$

where subscript $i$ indicates municipality, subscript $t$ indicates year, subscript BF indicates period before flowering, subscript AF indicates period after flowering, $y_{i,t}$ is the yield of buckwheat, $T_{i,t}$ is the averaged daily mean temperature, $P_{i,t}$ is the averaged daily mean precipitation, $S_{i,t}$ is the integrated solar radiation, and $\epsilon_{i,t}$ indicates the error term. We assumed a standard distribution as the error distribution. $f_{\mathrm{T}}$, $f_{\mathrm{P}}$, and $f_{\mathrm{S}}$ represent the spline functions for temperature, precipitation, and solar radiation, respectively. The subscript BF or AF is added because the spline function was estimated for each period (before flowering and after flowering), respectively. $f_{\mathrm{F}}$ represents the spline functions for the harvested area. $f_{\mathrm{t}}$ represents the spline functions for year. Using this model (2) as the full model, we selected the best model that has the lowest AIC value. All procedure was conducted by using the "gam" function of the mgcv package of R version 4.4.0 [33], in which the default setting was used.

## Prediction of the future yield

We predicted future buckwheat yields using the constructed generalized additive model and the growth rate model. First, we calculated the timing of flowering and harvesting for each municipality for every year (2020-2022, 2051–2060) using daily temperature under the future climate conditions and day length. Then, we calculated the future buckwheat yield with the model constructed using the procedure described in the previous section. We assumed that the harvested area for buckwheat would not change in the future: $f_F(F_{i,t})$ was fixed at that of 2020. The value of $f_t(t)$ was also fixed at $f_{\mathrm{t}}(2020)$ in the future prediciton. Therefore, the predicted future yields reflected the influence of the changes in meteorological values only. The crop yields during the period 2020-2022 were also predicted using the output of the five GCMs because the potential change in the yields was evaluated using the change rate (%) of the crop yield during 2051-2060 relative to that during 2020-2022.

## Language and packages used for analysis

All figures, including the maps, were created using R version 4.4.0 [33]. For combining two figures and adding annotations (e.g., a and b), PowerPoint for Mac version 16.83 was used. The libraries used for the analysis and figure generation were "latex2exp" for describing superscripts in the figures, "sf" for treating shape files, "dplyr" for manipulating the dataframe, "mgcv" for the analysis using the generalized additive model, "ggplot2" for making the plots, "fields" for displaying the heatmap in Fig 1, "tidyverse" for efficient coding, and "imager" for image processing.

## Results

### The relationship between growth rate and day length and temperature

Parameters of the model for the growth rate were estimated for each growing period, respectively (from sowing to flowering and from flowering to harvest). Table 1 shows the parameters estimated for each period. Pearson's coefficients of correlation between the estimated growing periods and the periods of the crop calendar were 0.41 (the period before flowering) and 0.32 (the period after flowering) (both $p$ values < 0.01).

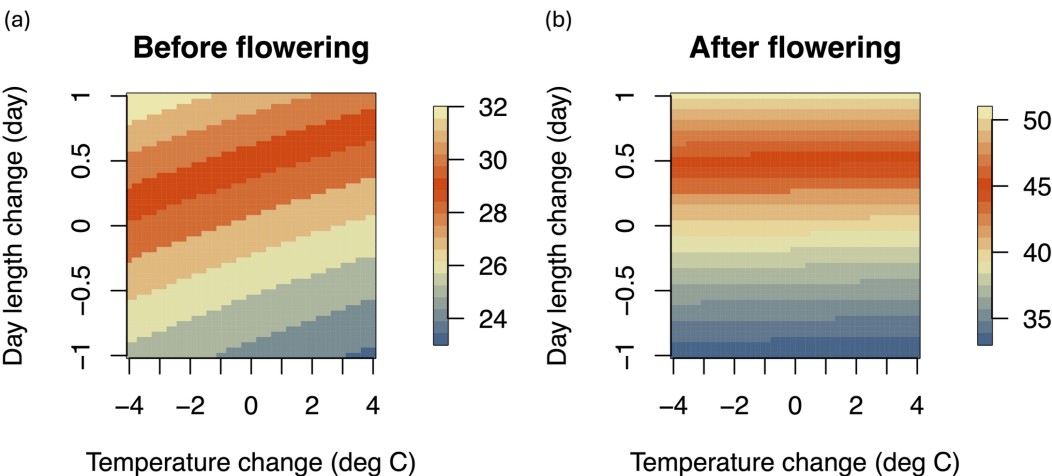

**Fig 1. Relationship between temperature, day length, and growing period.** The predicted length of the growing period before (a) and after (b) flowering is represented by color. The vertical axes indicate the change in the day length relative to the average values throughout Japan. The horizontal axes indicate the change in the temperature relative to the average values throughout Japan.

**Table 1. Estimated parameters of the model of growth rate.**

| Parameter | the period before flowering | the period after flowering |
|---|---|---|
| $A$ | $3.7 \times 10^{-2} \pm 1.1 \times 10^{-1}$ | $-2.4 \times 10^{-3} \pm 7.6 \times 10^{-4}$ |
| $B$ | $-7.0 \times 10^{-4} \pm 2.9 \times 10^{-4}$ | $1.3 \times 10^{-3} \pm 9.4 \times 10^{-6}$ |
| $L_c$ | $2.1 \times 10 \pm 1.9$ | $1.5 \times 10 \pm 4.0 \times 10^{-3}$ |
| $T_n$ | $-9.8 \times 10^{-1} \pm 1.1$ | $1.3 \pm 7.1 \times 10^{-3}$ |
| $G$ | $9.7 \pm 2.6 \times 10^{-1}$ | $1.0 \times 10 \pm 3.7 \times 10^{-3}$ |

Each value after ± represents the standard deviation.

Fig 1 shows the relationship between the length of the growing period before (left) and after flowering (right), and changes in temperature and day length according to the parameters estimated for the two models. During the period before flowering, as the temperature increases (horizontal axis), the growth rate increases, and the length of the period shortens. The day length had a large effect on the growth rate for both periods; as the day length increases (vertical axis), the growth rate decreases, and the length of the growing period is extended. According to the parameter values estimated, the temperature after flowering does not have an important effect on the growth rate.

## The relationship between climatic factors and yield

The model selection procedure by the AICs supported the full model (Eq. 2). Fig 2a shows the relationship between the integrated solar radiation (MJ m$^{-2}$) during the period before flowering and the yield response (kg 10 a$^{-1}$). Though there was a local optimal value at around 600 MJ m$^{-2}$, overall, the yield decreased according to the increase in the integrated solar radiation before flowering. Fig 2b shows the relationship between the integrated solar radiation (MJ m$^{-2}$) during the period after flowering and the yield response (kg 10 a$^{-1}$). Although there were slight increases and decreases, overall, the yield of buckwheat tended to increase according to the increase in the integrated solar radiation after flowering.

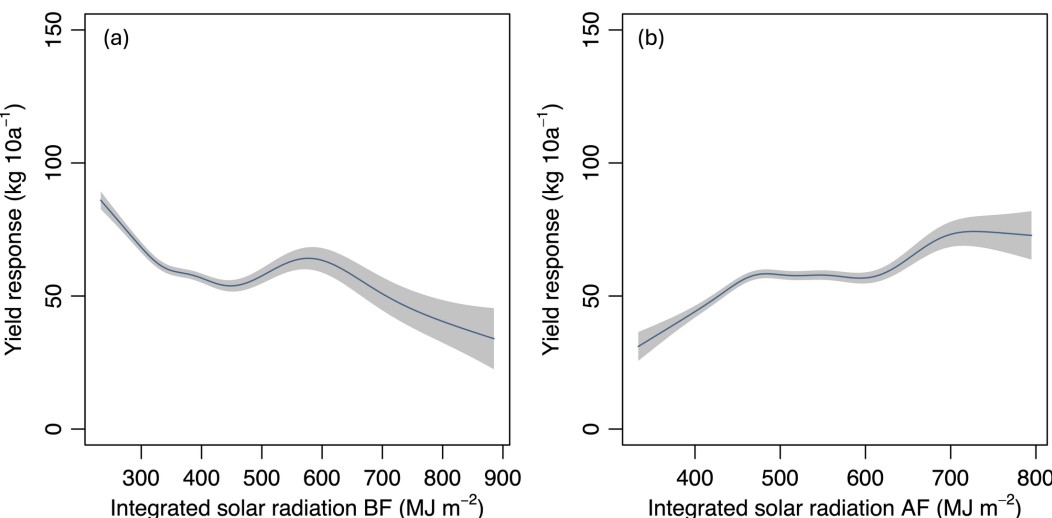

**Fig 2. Relationship between solar radiation and yield response.** The relationships between the integrated solar radiation (MJ m$^{-2}$) before flowering (a) or after flowering (b) and the yield (kg 10a$^{-1}$) response for buckwheat are represented. The integrated solar radiation means the sum of the values within the period before or after flowering (BF or AF). Areas shaded in gray indicate 95% confidence intervals.

S1 Fig shows the relationship between the mean temperature (°C) during the period before or after flowering and the yield response (kg 10 a$^{-1}$). The yield did not largely change with the change in the mean temperature before flowering, while the yield slightly decreased according to an increase in the mean temperature after flowering. S2 Fig shows the relationship between the mean precipitation (mm day$^{-1}$) during the period before or after flowering and the yield response (kg 10 a$^{-1}$). The mean precipitation before flowering had a local optimal value at around 15 mm day$^{-1}$, while the yield largely decreased with an increase in the mean precipitation after flowering between 0 and 4 mm day$^{-1}$.

The coefficient of determination ($R^2$) between the estimated yields and the observed yields was 0.23. The root mean squared error (RMSE) was 29.77 kg 10a$^{-1}$ (the average yield observed was 65.57 kg 10a$^{-1}$). While the estimated $R^2$ is low, this is due to our deliberate choice of a simple model. We analyzed nationwide data from Japan to examine the relationship between meteorological factors and yield by estimating a single spline function for the entire country, an approach adopted to mitigate extrapolation issues in future predictions. Although one might expect that using distinct spline functions for each municipality would yield a higher $R^2$, such modifications would likely introduce extrapolation problems when future temperatures exceed the range of our training data. Thus, the model's simplicity and limited flexibility naturally result in a lower $R^2$. Nonetheless, the detection of a statistically significant relationship between meteorological factors and yield with this model attests to the robustness of the relationship. The main target of this study is definitely not to make a high-precision model. The purpose of this study is to investigate how increased temperature affects crop yields through changes in the growing season and untangle the complex relationship between the changes in temperature, day length, solar radiation, and growing rate of the crop. Moreover, making models using machine-learning methods should not fit the purpose of this study, even if such methods slightly increase the precision accuracy of the model.

## Future prediction of growing period

The lengths of the period before and after flowering under the future climate condition (2051–2060) were predicted using the model (Eq. 1) with the parameter values estimated. Fig 3a shows the predicted differences in the length (days) of the period before flowering for 2051–2060 compared to 2020–2022. The result shows that temperature increases due to climate change increase the growth rate and shorten the period before flowering. On average, the length of the period before flowering was predicted to be shortened by $0.41 \pm 0.14$ days (after $\pm$ is the standard deviation, same below). Fig 3b shows the predicted differences in the length (days) of the period after flowering for 2051–2060 compared to 2020–2022. On average, the length of the period after flowering was predicted to be slightly lengthened by $0.075 \pm 0.17$ days.

## Future predicted changes of climate factors

Fig 4a shows the predicted change rate (%) in the integrated solar radiation before flowering for 2051–2060 compared to 2020–2022, including the effect of the change in the growing periods due to temperature change. On average, the integrated solar radiation was predicted to increase by $3.85 \pm 2.77$ %. Fig 4b shows the predicted change rate in the integrated solar radiation after flowering for 2051–2060 compared to 2020–2022, including the effect of the change of the growing period due to temperature change. On average, the integrated solar radiation after flowering was predicted to increase by $6.55 \pm 3.71$ %. Figs S3 and S4 show the

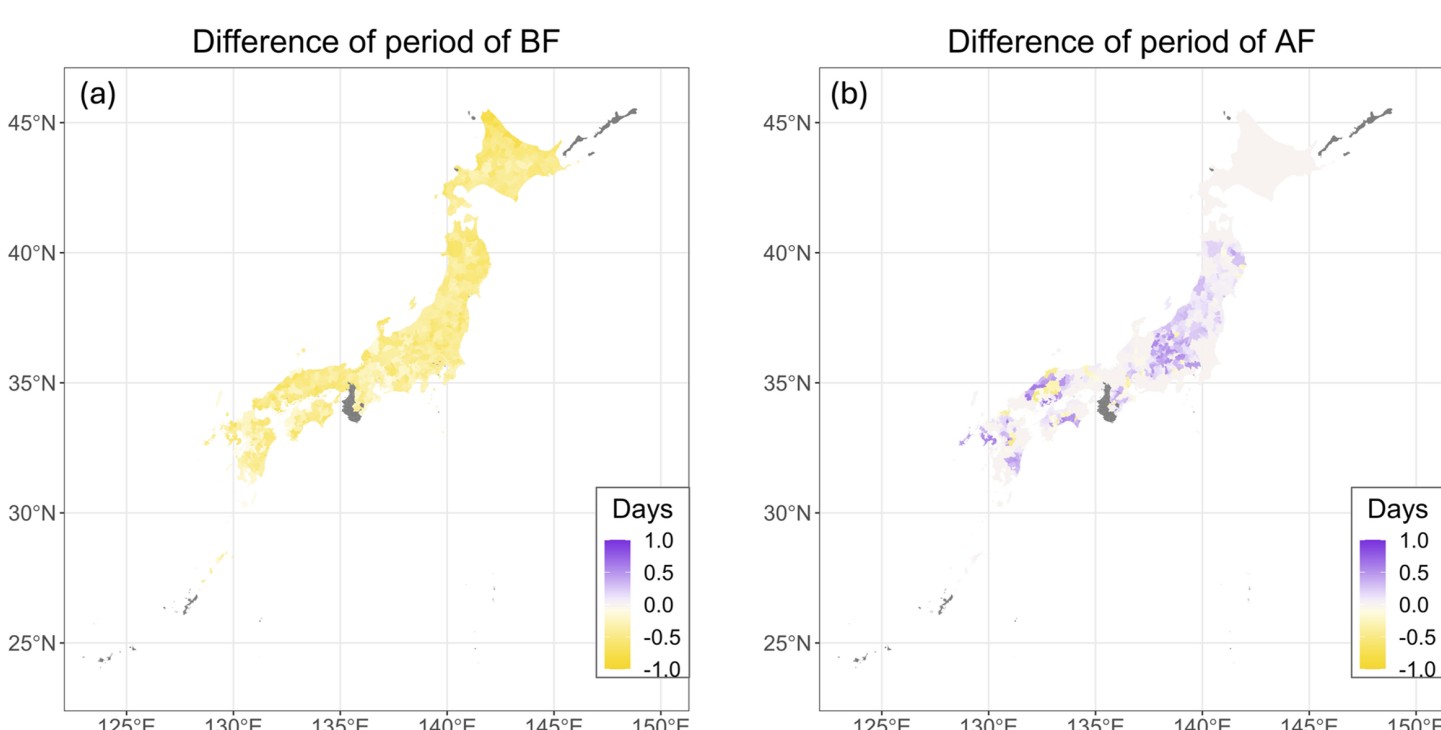

**Fig 3. Maps of the differences of the growing periods between future and current conditions.** The distribution of the prediction of the difference in the length (days) of the periods before (a) and after (b) flowering for 2051–2060 compared to 2020–2022 under the RCP8.5–SSP5 scenario is mapped. The average value of the results predicted by the five GCMs is described. Areas with no data are shown in gray. The colors are divided by municipalities. The map is based on the shapefile from the National Land Numerical Information Download Site (CC BY 4.0).

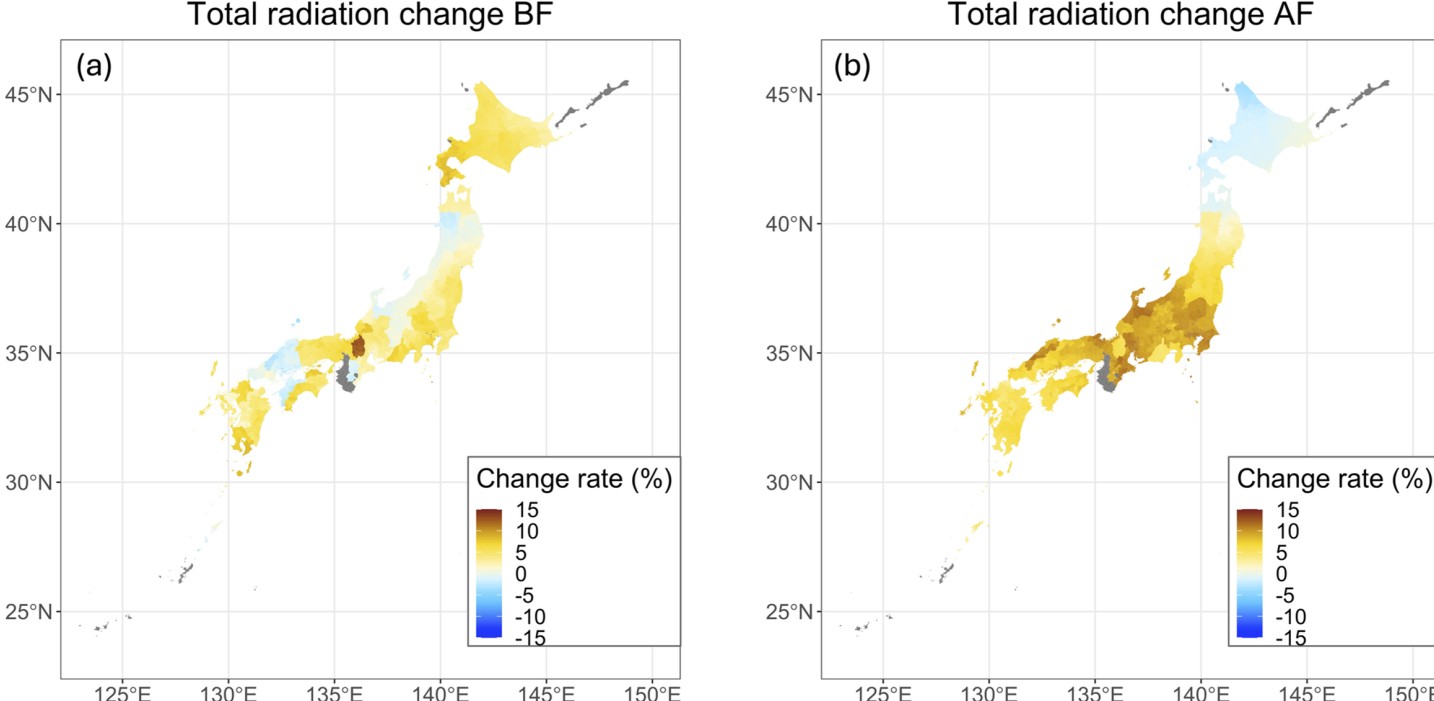

**Fig 4. Maps of the change rate of the integrated solar radiation.** Predicted change rates (%) in the mean solar radiation during the period before flowering (a) and those in the integrated solar radiation during the period after flowering (b) for 2051–2060 compared to 2020–2022 under the RCP8.5-SSP5 scenario are mapped. Note that the effects of the change in the growing periods due to temperature change on the integrated solar radiation were also included in this prediction. The mean values of the results predicted by the five GCMs are depicted. Areas with no data are shown in gray. The colors are divided by municipalities. The map is based on the shapefile from the National Land Numerical Information Download Site (CC BY 4.0).

change rate (%) in the mean temperature and the mean precipitation before and after flowering, respectively. Table 2 summarizes the predicted change rates of the climatic factors for 2051–2060 compared to 2020–2022. On average, the mean temperature and the mean precipitation before and after flowering were predicted to increase. However, for precipitation, the predicted change rates varied widely throughout Japan.

## Difference in the change rate in climate factors

We evaluated the difference in the change rate in climate factors between the prediction in which the change of the phenology was considered and that in which the phenology was fixed at the current condition. Fig 5a shows differences in the integrated solar radiation predicted

**Table 2. Estimated difference in meteorological factors between future and current conditions.**

| Meteorological values | Before flowering | After flowering |
|---|---|---|
| Mean temperature | 6.82 ± 0.99 (%) | 10.32 ± 1.44 (%) |
| Mean precipitation | 6.15 ± 22.93 (%) | 8.12 ± 14.95 (%) |
| Integrated solar radiation | 5.34 ± 2.80 (%) | 3.82 ± 2.76 (%) |

Summary of the predicted change rate (%) in the meteorological values for 2051–2060 compared to 2020–2022 under the RCP8.5-SSP5 scenario considering the effect of the change of the growing periods before and after flowering. Each value after ± represents the standard deviation.

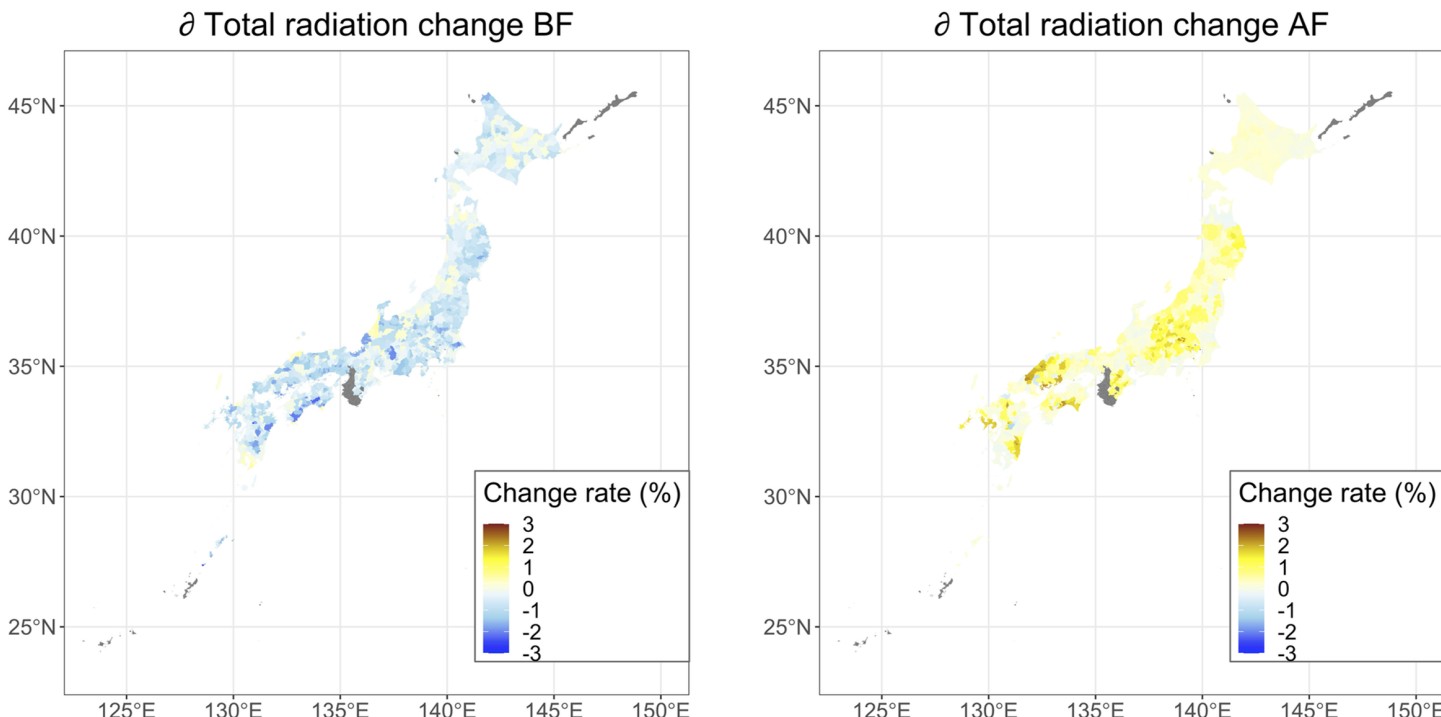

**Fig 5. Maps of the impact of the change of growing period on solar radiation.** The differences in the integrated solar radiation predicted between the condition in which the growing periods were changed according to future temperature change and that in which the growing periods were fixed (the same as the average in 2020-2022) are mapped. The average value of the results predicted by each of the output values of five GCMs is mapped. Areas with no data are shown in gray. The colors are divided by municipalities. The map is based on the shapefile from the National Land Numerical Information Download Site (CC BY 4.0).

between the condition in which the growing periods were changed according to future temperature change and that in which the growing periods were fixed (the same as the average in 2020-2022). According to the shortened period before flowering, the predicted integrated solar radiation decreased during 2051-2061. On the other hand, the predicted integrated solar radiation after flowering increased during 2051-2061 because of the elongated period between flowering and harvesting (Fig 5b).

## Future prediction of the yield

Fig 6 shows the predicted change rates (%) in the future buckwheat yields for 2051–2060 compared to 2020–2022. The yields were predicted to increase in the mid and Western areas, while decreases were expected in the Eastern region. On average, buckwheat yield was predicted to increase by 10.22 ± 19.26 %.

We evaluated the effect of the shift of the flowering time in the future climate conditions on buckwheat yield through the change in solar radiation. In assessing the effect, we compared the yields under future climate conditions with those under future climate conditions in which only solar radiation was calculated with the current growing periods. In the latter condition, the integrated solar radiation was calculated using the flowering time that was expected under the current climate condition (2020-2022) (we call the condition "fixed radiation prediction" hereafter). By comparing these two settings, we can evaluate the effect of the change in integrated solar radiation only through phenology change on the yields. Figs 7a and b show the difference in the estimated change rate of the buckwheat yields (yield

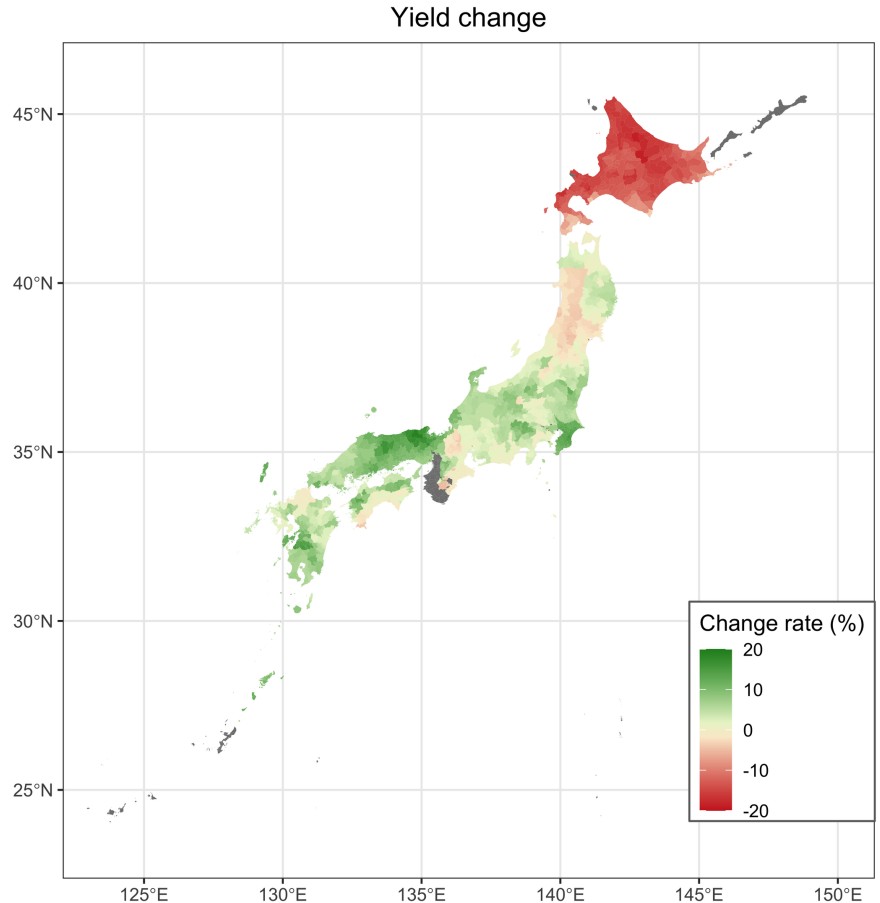

**Fig 6. Predicted yield change of buckwheat.** Predicted change rates (%) in the future buckwheat yield for 2051–2060 compared to 2020–2022 under the RCP8.5-SSP5 scenario is mapped. The mean values of the results predicted by the five GCMs are described. Areas with no data are shown in gray. The colors are divided by municipalities. The map is based on the shapefile from the National Land Numerical Information Download Site (CC BY 4.0).

change of 2051-2060 relative to 2020-2022) between the normal prediction and the fixed radiation prediction. Before flowering, the positive effects of the shortening of the growing period on yields were estimated in the western region. In contrast, the negative impacts on yields were calculated in some of the eastern areas. This result would reflect the nonlinear relationship between integrated solar radiation and buckwheat yield before flowering (see Fig 2a).On the other hand, the positive effects of the elongated reproductive period on yields were estimated all over Japan for the period after flowering. This result would reflect the positive relationship between integrated solar radiation and yield after flowering (see Fig 2b).

## Discussion

This study evaluated the complex relationships between climate change, the growing period, and the crop yield through statistical analyses to assess the effects of future climate change. First, using historical data, we analyzed the relationship between buckwheat yield and climatic factors using the generalized additive model before and after flowering, respectively. Second, using crop calendar data, the model parameters for the growth rate of buckwheat

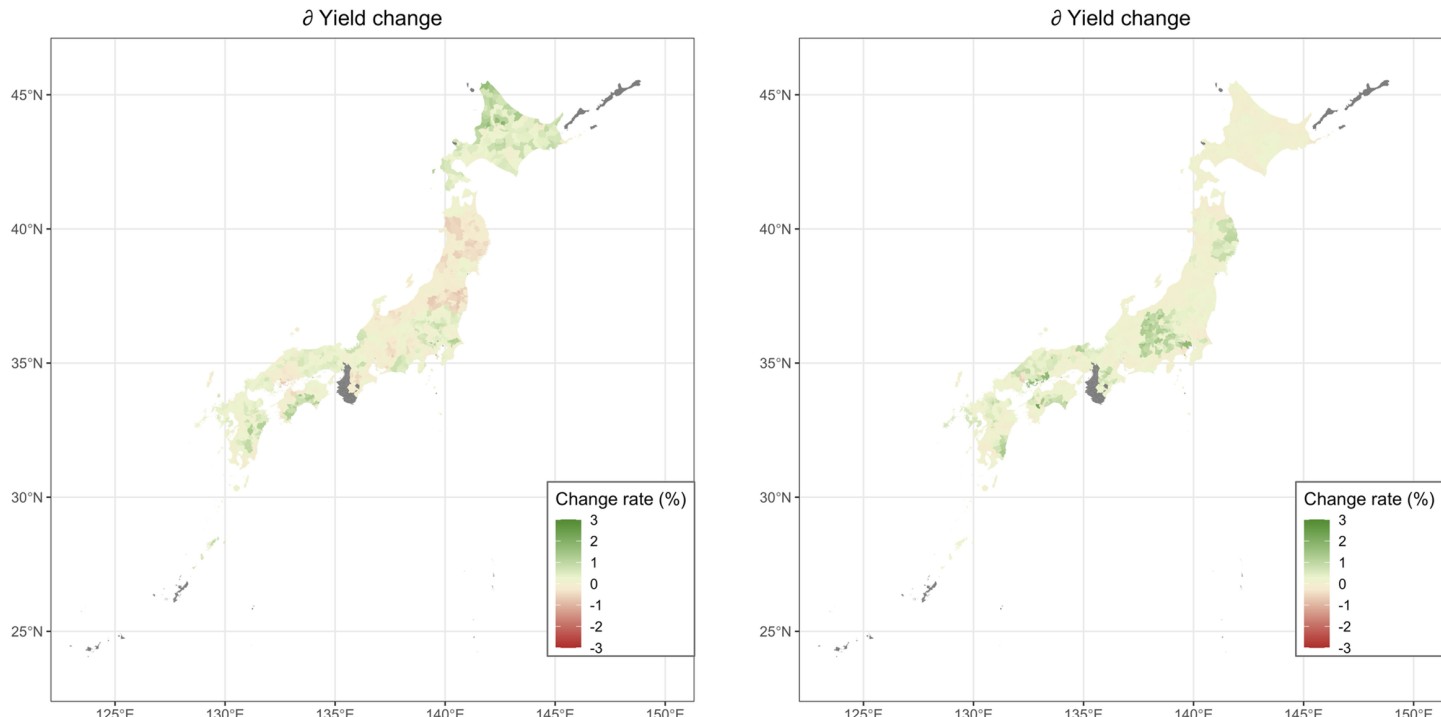

**Fig 7. Maps of the impact of the change of growing period on buckwheat yields through solar radiation change.** The differences in the buckwheat yields predicted between the condition in which the growing periods were changed according to future temperature change and that in which the growing periods were fixed (the same as the average in 2020-2022) are mapped. Note that, in the latter condition, only the integrated solar radiation is changed to isolate the effect of the change of the integrated solar radiation on the yield. The average value of the results predicted by each of the output values of five GCMs is mapped. Areas with no data are shown in gray. The colors are divided by municipalities. The map is based on the shapefile from the National Land Numerical Information Download Site (CC BY 4.0).

were estimated to determine the relationship between the temperature, the day length, and the growth rate. Finally, the potential yields under future climate conditions were predicted.

For the relationship between the yield and solar radiation, it is suggested that the integrated solar radiation positively affected the yield after flowering (Fig 2). A previous study [35] reported that the solar radiation after flowering led to an increase in the number of insects visiting buckwheat. They also reported that the increased precipitation led to a decrease in insects. The estimated relationship in this study between the precipitation or the integrated solar radiation and the yield response had the same tendency after flowering that is reported by these previous studies. However, the mean precipitation before flowering did not clearly affect the yields in this study. It was reported that buckwheat is particularly susceptible to high soil moisture at the sowing stage [36,37]. Therefore, the yield is expected to show a negative response to increased mean precipitation before flowering. According to the crop calendar data used in this study, the average period from sowing to germination was 7.37 ± 5.03 days. On the other hand, the length of the period before flowering was 37.27 ± 6.13 days. Therefore, it is possible that the average precipitation during the period could not catch the sensitive period in which soil conditions largely affect crop growth.

For the relationship between the yield and temperature, during the period before flowering, the mean temperature had an optimum temperature at around 18 °C in the range of 15-20 °C (S1 Fig a). On the other hand, a slightly positive effect was estimated in the range above 20 °C (S1 Fig a). According to a previous study [38], high-temperature conditions

(27 °C) do not largely affect the vegetative growth of the buckwheat relative to the middle-temperature conditions (21 °C) in the growth chamber condition. On the other hand, another study suggest that a temperature of 30°C, compared to 20°C, is actually more favorable for the vegetative growth of common buckwheat via enhanced photosynthesis rate [39]. Therefore, the slightly positive correlation between yield and temperatures above 20°C obtained in our analysis does not contradict the suggestions of previous studies. In contrast, during the reproduction stage, the previous study suggests that the high-temperature conditions had several negative effects, such as the rates of seed and flower abortions [38]. Our analysis showed that, after flowering, temperature exerts a slightly negative effect on yield (S1 Fig b). Although the estimated relationships between the yield and temperature in this study did not contradict the result of the previous study, it would not be easy to interpret the relationship between the yield and temperature obtained by the analysis, particularly during the period before flowering because the estimated yield response once decreased in the middle-temperature range (18-20 °C). Further analysis using the data that includes the yield data at this range may elaborate on the relationship between yield and temperature in future studies. In summary, the estimated relationship between yield response and climatic factors in this study did not basically contradict the results of the previous studies. This implies that even the statistical analysis using the past climate and yield data can potentially evaluate the biological characteristics of the crops.

Using the crop calendar data to estimate the parameters of the models for the growth rate, it was suggested that, before flowering, a temperature increase enhanced the growth rate, and an increase in day length inhibited the growth rate. Also, an increase in day length inhibited the growth rate after flowering, but the temperature did not largely affect the growth rate. In the previous studies, Michiyama and Sakurai [40] showed that when buckwheat was cultivated under three conditions (19–15 °C (at day time–at night time), 19–25 °C and 30–24 °C), the flowering time became earlier as the temperature increased. Lachmann et al. [41] also showed that the flowering time became earlier when cultivated at 20-25 °C compared to when cultivated at a lower temperature condition. On the other hand, [38] showed that, when two buckwheat varieties were cultivated at 27 °C, the flowering time became 6–13% later than when those were cultivated at 21 °C. Although it may be difficult to directly compare the results obtained in this study with the results obtained in the previous studies because the experiment settings differed, previous studies have not had concordant results about the effect of temperature on the growth rate. On the other hand, previous studies examining the relationship between day length and the flowering time showed that long day length treatment delayed the flowering time [41,42]. These results are consistent with our results.

The yield of buckwheat in 2051–2060 predicted by the generalized additive model and the growth rate model increased by 10.22 ± 19.26 % on average across Japan compared to the yield in 2020–2022 expected by the model (Fig 6). Focusing on the meteorological values incorporated into the model, the integrated solar radiation during the period before flowering increased by 4.66 ± 3.21 %, and the integrated solar radiation during the period after flowering increased by 5.62 ± 3.30 % (Table 2, Fig 4). Similar to the solar radiation, the mean temperature (before flowering: 5.75 ± 1.22 %, after flowering: 9.52 ± 1.25 %) and the mean precipitation (before flowering: 12.34 ± 27.56 %, after flowering: 10.35 ± 19.79 %) are also predicted to increase across Japan (Table 2, Figs S3 and S4), but compared to the solar radiation and the mean temperature, the mean precipitation had a larger variation among areas and is predicted to decrease in some areas, especially during the period before flowering (Fig S4).

In the prediction during 2051–2060, the length of the period before flowering was shortened due to increased temperature. This resulted in a slightly longer day length after flowering, which led to a decreased growth rate after flowering (Fig 5). The extension in the

length of the period after flowering due to reduced growth rate can increase the integrated solar radiation at the period after flowering, which contributed to the increase of the yields in many regions. That is, increased temperatures indirectly enhance buckwheat yield through the shortening of the pre-flowering period. In fact, our study shows that an increase in post-flowering temperature directly has a slightly negative impact on yield (S1 Fig b). However, our results indicate that rising temperatures lead to a shortened pre-flowering period, which in turn results in increased day length after flowering, thereby extending the post-flowering period, increasing the cumulative solar radiation after flowering, and ultimately enhancing yield. The fact that this yield increase outweighs the negative effects of higher post-flowering temperatures is the key factor behind the estimated yield enhancement due to increasing temperatures.

In general, it has been suggested that, as temperatures increase, the growth rate of the crop increases, thus shortening the time to flowering and maturity. In process-based crop models, the rate of photosynthesis is largely determined by the amount of total solar radiation on any given day, and the daily photosynthetic products produced by photosynthesis are converted to crop biomass, which accumulates throughout the growing period. Thus, the shorter the growing period, the less photosynthetic product is accumulated, and the smaller the final crop biomass. For these reasons, in crop models, increased temperatures often have a negative impact on the crop yields [16–22]. However, the results of this study suggest that the changes in the growing period would not be so simple, as they are affected by various factors, including changes in temperature, solar radiation, and day length. In the study by Kolaric et al. [43], although it was not statistically significant, a slightly positive correlation between buckwheat yield and temperature was reported. Our result is consistent with the report by Kolaric et al. and further demonstrates the mechanism through which yield increases.

In summary, the analyses of this study did not show a simple relationship in which a shorter growing period leads to a decrease in the integrated solar radiation and a decrease in the yield. It was suggested that the period after flowering will become longer as the period before flowering will become shorter in future climate conditions. As a result, the opposite of what had previously been thought to have occurred was that a reduction in the period before flowering had a positive effect on the yield.

The prediction accuracy of the model of this study was not high ($R^2 = 0.23$). There are two possible reasons in addition to the reason rised in the result section for this low prediction accuracy. First, the model was a simple statistical model in which the averaged climatic factors were included as the explanatory variables. However, the crops should respond to daily climatic conditions. Therefore, the effects of climatic factors should be underestimated when the average climatic values during cropping seasons are used. Second, the factors that affect crop yields should not be only climatic factors. The soil condition, the crop varieties, and slight differences in farming methods among municipalities should largely affect the crop yield. If we adopt one of the machine learning methods that includes all these factors as the feature values, we may obtain a model with higher prediction accuracy. However, the main purpose of this study is not to make a high-accuracy prediction model of buckwheat yield. The target of this study is to abstract the robust climatic effect on the yield and untangle the complex relationship between the changes in temperature, day length, solar radiation, and the growing rate of the crop. Therefore, a more robust approach, such as that used in this study, would be preferable to analyze the effects clearly.

Moreover, the analyses in this study do not consider the effect of carbon dioxide fertilization. Buckwheat is a C3 plant, and the fertilization effect of carbon dioxide is considered to

have a positive impact [44]. However, even considering this, the present conclusion does not change.

The results of this study were analyzed for common buckwheat in Japan only. Therefore, of course, different results may be obtained for other crops and other regions. However, this study suggests that the decrease in the growing period due to temperature increases does not simply result in a reduction in the yield due to a decrease in the integrated solar radiation. This study used statistical analyses to analyze the relationship between the yield and meteorological factors, such as solar radiation. Then, it examined how changes in the growing period will affect it in the future. This is a different approach from the previous studies. The complex effects that changes in growing periods have on future yields will become clearer through statistical analysis using historical data in future studies.

## Conclusion

In this study, first, we constructed a generalized additive model for Japanese common buckwheat using yield data and past meteorological data and clarified the relationship between buckwheat yield and meteorological factors. As a result, it was suggested that the effects of the temperature, the precipitation, and the solar radiation on buckwheat yield differ between the period before flowering and the period after flowering. In particular, regarding solar radiation, there was a negative effect of the increase of integrated solar radiation on the buckwheat yields before flowering, whereas there was a tendency for yield to increase in proportion to the integrated solar radiation during the period after flowering. Second, using the crop calendar data, we estimated the parameters of the model for the growth rate and analyzed the relationship between the growing period (before or after flowering), the temperature, and the day length. The results suggested that the length of the period before flowering was shortened by the increase in temperature, whereas the increase in the day length extended the respective length (days) of both the periods before and after flowering. Third, the results obtained above were used to predict the future growing periods and the future yields of buckwheat. It was predicted that the temperature increases associated with climate change in future scenarios led to the reduction of the period before flowering but a slight increase in the day length after flowering, which extends the growing period after flowering. As a result, the shift of flowering time resulted in an increase in the buckwheat yield predicted in future climate conditions. This study shows that changes in the growth rate of buckwheat due to increased temperature could positively increase the yields through increased day length after flowering through statistical analysis using past crop data. This study shows that changes in growth rate due to increased temperature could have complex effects on crop yield. Further research using past crop data should evaluate the effects of increasing temperature on crop phenology and its impact on yield.

## Supporting information

**S1 Fig. Relationship between the mean temperature (°C) during the period before flowering (a) or after flowering (b) and the yield response for buckwheat.** The mean temperature was averaged over the period before or after flowering. Areas shaded in gray indicate 95% confidence intervals.
(TIF)

**S2 Fig. Relationship between the mean precipitation (mm day$^{-1}$) during the period before flowering (a) or after flowering (b) flowering and the yield response for buckwheat.** The

mean precipitation was averaged over the period before or after flowering. Areas shaded in gray indicate 95% confidence intervals.
(TIF)

**S3 Fig. Predicted change rates (%) in the mean temperature during the period before flowering (a) or after flowering (b) in 2051–2060 relative to 2020–2022 under the RCP8.5-SSP5 scenario.** The mean values of the results predicted by the five GCMs are depicted. Areas with no data are shown in gray. The colors are divided by municipalities. The map is based on a shapefile from the National Land Numerical Information Download Site (CC BY 4.0).
(TIF)

**S4 Fig. Predicted change rates (%) in the mean precipitation during the period before flowering (a) or after flowering (b) for 2051–2060 relative to 2020–2022 under the RCP8.5-SSP5 scenario.** The mean values of the results predicted by the five GCMs are depicted. Areas with no data are shown in gray. The colors are divided by municipalities. The map is based on a shapefile from the National Land Numerical Information Download Site (CC BY 4.0).
(TIF)

## Acknowledgments

I am grateful to the members of the Agro-Environmental Informatics Group, NARO for their helpful discussions and supports.

## Author contributions

**Conceptualization:** Gen Sakurai.

**Data curation:** Gen Sakurai, Norikazu Okabe.

**Formal analysis:** Gen Sakurai.

**Funding acquisition:** Gen Sakurai.

**Investigation:** Gen Sakurai.

**Methodology:** Gen Sakurai.

**Project administration:** Gen Sakurai.

**Validation:** Gen Sakurai.

**Visualization:** Gen Sakurai.

**Writing – original draft:** Gen Sakurai, Naoki Ishitsuka.

**Writing – review & editing:** Naoki Ishitsuka, Norikazu Okabe.

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
