## [Decision Letter · Decision Letter 0]

22 Jul 2024

PONE-D-24-13382Complex relationship between crop yields and crop growing period: The shortened growing period before flowering contributes to yield increase in buckwheatPLOS ONE

Dear Dr. Sakurai,

Thank you for submitting your manuscript to PLOS ONE. After careful consideration, we feel that it has merit but does not fully meet PLOS ONE’s publication criteria as it currently stands. Therefore, we invite you to submit a revised version of the manuscript that addresses the points raised during the review process.

The current manuscript is written scientifically, and it has valuable findings. Only a few general suggestions are provided.

• All the libraries used in the software environment should be mentioned in the text.

• What method was used for interpolation in the maps?

• Regarding the precision and accuracy of the fitted model, statistical information (R-squared, RMSE, Index of Agreement, . . .) is needed to compare the observed values and the simulated values.Please ensure that your decision is justified on PLOS ONE’s publication criteria and not, for example, on novelty or perceived impact.

We look forward to receiving your revised manuscript.

Kind regards,

Juan Carlos Suárez Salazar

Academic Editor

PLOS ONE

“This research was performed by the Environment Research and Technology 397 Development Funds S18 and 2G-2201 (JPMEERF20S11803 and JPMEERF20222G01) 398 of the Environmental Restoration and Conservation Agency provided by the Ministry of 399 the Environment of Japan.”

“This research was performed by the Environment Research and Technology Development Funds S18 and 2G-2201 (JPMEERF20S11803 and JPMEERF20222G01) of the Environmental Restoration and Conservation Agency provided by the Ministry of the Environment of Japan.”

“This research was performed by the Environment Research and Technology 397 Development Funds S18 and 2G-2201 (JPMEERF20S11803 and JPMEERF20222G01) 398 of the Environmental Restoration and Conservation Agency provided by the Ministry of 399 the Environment of Japan.”

5. We note that Figures 3, 4, 5, 6, 7, S3 and S4 in your submission contain [map/satellite] images which may be copyrighted. All PLOS content is published under the Creative Commons Attribution License (CC BY 4.0), which means that the manuscript, images, and Supporting Information files will be freely available online, and any third party is permitted to access, download, copy, distribute, and use these materials in any way, even commercially, with proper attribution. For these reasons, we cannot publish previously copyrighted maps or satellite images created using proprietary data, such as Google software (Google Maps, Street View, and Earth). For more information, see our copyright guidelines: http://journals.plos.org/plosone/s/licenses-and-copyright.

1. You may seek permission from the original copyright holder of Figures 3, 4, 5, 6, 7, S3 and S4 to publish the content specifically under the CC BY 4.0 license. 

Additional Editor Comments:

The current manuscript is written scientifically, and it has valuable findings. Only a few general suggestions are provided.

• All the libraries used in the software environment should be mentioned in the text.

• What method was used for interpolation in the maps?

• Regarding the precision and accuracy of the fitted model, statistical information (R-squared, RMSE, Index of Agreement, . . .) is needed to compare the observed values and the simulated values.

Reviewers' comments:

Reviewer's Responses to Questions

**Comments to the Author**

1. Is the manuscript technically sound, and do the data support the conclusions?

Reviewer #1: Yes

2. Has the statistical analysis been performed appropriately and rigorously? 

Reviewer #1: Yes

3. Have the authors made all data underlying the findings in their manuscript fully available?

Reviewer #1: Yes

4. Is the manuscript presented in an intelligible fashion and written in standard English?

Reviewer #1: Yes

5. Review Comments to the Author

Reviewer #1: First, I would like to take a moment to express my deepest gratitude for inviting me to review the manuscript titled “Complex relationship between crop yields and crop growing period: The shortened growing period before flowering contributes to yield increase in buckwheat”.

The manuscript is about one of the most important issues, climate change and crop yields. This subject has many interested people today and the results of this research are useful for predicting the crop yields in different regions. The current manuscript is written scientifically, and it has valuable findings. Only a few general suggestions are provided.

• All the libraries used in the software environment should be mentioned in the text.

• What method was used for interpolation in the maps?

• Regarding the precision and accuracy of the fitted model, statistical information (R-squared, RMSE, Index of Agreement, . . .) is needed to compare the observed values and the simulated values.

6. PLOS authors have the option to publish the peer review history of their article (what does this mean?). If published, this will include your full peer review and any attached files.

Reviewer #1: **Yes: **Hedayatollah Karimzadeh Soureshjani

---

## [Author Response · Author response to Decision Letter 1]

22 Nov 2024

The authors express their sincere gratitude to the reviewers for their time and effort in providing valuable feedback. Their insightful comments and suggestions have significantly contributed to enhancing the quality of this manuscript. In response to the reviewers' comments, the manuscript has been revised as follows: reviewers' comments are presented in bold italics, while the authors' responses are presented in Roman type. In the "Revised Manuscript with Track Changes" document, deletions are indicated by strikethroughs in red, and additions are indicated by blue font.

All the libraries used in the software environment should be mentioned in the text.

The authors appreciate the reviewers' insightful comments. In response to the reviewers' suggestions, a new section entitled "Language and Packages Used for Analysis" has been added before the Results section to provide further detail regarding the analytical tools employed in this study.

What method was used for interpolation in the maps?

In the figures, no interpolation method was used. The colors of the map are divided by municipalities. The total number of municipalities is more than 1,000. Therefore, while the map appears to be interpolating within the map, the colors are actually assigned to each municipality. This is also indicated in the figure legends.

Regarding the precision and accuracy of the fitted model, statistical information (R-squared, RMSE, Index of Agreement, . . .) is needed to compare the observed values and the simulated values. Please ensure that your decision is justified on PLOS ONE’s publication criteria and not, for example, on novelty or perceived impact.

Thank you for the reviewer's valuable feedback. We have added statistics on the accuracy between the observed and estimated values of the yield data in the Results section. We have also discussed the values of these statistics in the Discussion section.

---

## [Decision Letter · Decision Letter 1]

22 Jan 2025

PONE-D-24-13382R1Complex relationship between crop yields and crop growing period: The shortened growing period before flowering contributes to yield increase in buckwheatPLOS ONE

Dear Dr. Sakurai,

Thank you for submitting your manuscript to PLOS ONE. After careful consideration, we feel that it has merit but does not fully meet PLOS ONE’s publication criteria as it currently stands. Therefore, we invite you to submit a revised version of the manuscript that addresses the points raised during the review process. Please submit your revised manuscript by Mar 08 2025 11:59PM. If you will need more time than this to complete your revisions, please reply to this message or contact the journal office at plosone@plos.org. Please include the following items when submitting your revised manuscript:

We look forward to receiving your revised manuscript.

Kind regards,

Karthikeyan Thiyagarajan PhD

Academic Editor

PLOS ONE

Journal Requirements:

Additional Editor Comments:

Dear Authors,

On behalf of the Editorial Board, we thank you for submitting your manuscript to PLOS ONE. Reviewers have now completed their reviews of your manuscript, and one of the reviewers suggested a minor revision.

Please carefully read the comments and suggestions from reviewers and respond to each comment and revise the manuscript accordingly. Please submit the revised manuscript with both marked and unmarked versions by 12th February 2025.

I have following suggestions and comments:

I appreciate your work on climatic predictions, especially the effect of temperature on buckwheat yield, and you have examined the solar radiation after flowering predicted to have a positive impact on yield.

You have shown increased temperature with a positive impact on yield is questionable as rising temperature may cause terminal heat stress, etc. So clarification is needed on how increased temperature would

certainly have yield-related benefits while not having stress-related defects on crop productivity? Hence, please add some points concerning the temperature-induced heat stress-related consequences.

Was the model you have used linear or a linear mixed model? What kind of dependent and predictor variables used for the model?

Have you assigned the fixed effects of predictor variables?

Please mention the average buckwheat yield per acre in 1994, 2001, and for any recent year, as you have mentioned the number of sample sizes only and not the yield.

Why was the coefficient of determination (R²) weakly positive with a mere 0.23 of observed and estimated yields?

There are typos, for instance in line 57, the recent IPCC report [3]... Please increase the resolution of the figures.

Reviewers' comments:

Reviewer's Responses to Questions

**Comments to the Author**

1. If the authors have adequately addressed your comments raised in a previous round of review and you feel that this manuscript is now acceptable for publication, you may indicate that here to bypass the “Comments to the Author” section, enter your conflict of interest statement in the “Confidential to Editor” section, and submit your "Accept" recommendation.

Reviewer #1: All comments have been addressed

Reviewer #2: All comments have been addressed

Reviewer #3: All comments have been addressed

2. Is the manuscript technically sound, and do the data support the conclusions?

Reviewer #1: Yes

Reviewer #2: No

Reviewer #3: Yes

3. Has the statistical analysis been performed appropriately and rigorously? 

Reviewer #1: Yes

Reviewer #2: No

Reviewer #3: Yes

4. Have the authors made all data underlying the findings in their manuscript fully available?

Reviewer #1: Yes

Reviewer #2: Yes

Reviewer #3: Yes

5. Is the manuscript presented in an intelligible fashion and written in standard English?

Reviewer #1: Yes

Reviewer #2: No

Reviewer #3: Yes

6. Review Comments to the Author

Reviewer #1: (No Response)

Reviewer #2: This study analyzed how buckwheat yields are affected by temperatures and day length based on long-term data from all over Japan. The results indicated that rising temperatures speed up development and increase day length in the flowering period, thus increasing yield as a result. However, because the analysis is too simple, there are various problems and the results are unreliable.

The authors state the advantages of a simple model, but it is an oversimplification to state that the analysis without confounding factors would still produce qualitatively reliable results.

For example, buckwheat varieties are geographically unevenly distributed, which may create a spurious correlation with temperature. Also, any temporal trends in fertilizer application would affect yields, which might also show a spurious correlation with rising temperature in these decades. Furthermore, this study only deals with phenology and yield as crop data, but buckwheat plant size (biomass) is highly important. It is possible that higher temperatures could have increased plant size in a short term, resulting in higher yields. This surely have nothing to do with the effect of daylength after flowering.

If you really want to know the effects of temperature and day-length on buckwheat growth and yield, experiments under controlled environments will be able to uncover the cause-and-effect relationship, not by analyzing data that is full of uncertainties.

Reviewer #3: I would like to thank the authors for their effort in conducting this valuable study on buckwheat, a crucial crop with significant nutritional and agricultural benefits. Buckwheat plays a vital role in food security, especially under climate change conditions, and understanding its phenological responses to environmental factors is essential for optimizing yield. This research provides meaningful insights into how climate variability influences buckwheat productivity, which is critical for future agricultural planning and adaptation strategies.

7. PLOS authors have the option to publish the peer review history of their article (what does this mean?). If published, this will include your full peer review and any attached files.

Reviewer #1: **Yes: **Hedayatollah Karimzadeh Soueshjani

Reviewer #2: No

Reviewer #3: **Yes: **Mohamed M. Hassona

---

## [Author Response · Author response to Decision Letter 2]

10 Feb 2025

Response to the editor

We would like to express our sincere gratitude to the editor for dedicating their time and expertise to providing insightful feedback on our manuscript. We deeply appreciate your careful consideration and constructive comments, which have been invaluable in improving the quality of our work. In the following, we provide our responses to the editor's comments.

It is unfortunate that we received negative comments from Reviewer 2. However, we cannot agree with Reviewer 2’s comments. As we have addressed in our response to Reviewer 2, the criticisms raised represent a general indictment of all statistical analyses using crop data and stem from the reviewer’s belief that experimental research is the most superior approach. In reality, experimental studies and studies employing statistical big data should be seen as complementary, not as one being superior to the other. Although statistical data inherently contain significant errors, appropriate statistical analysis can extract valuable information, and in our study we have taken the utmost care to ensure that our analyses are free from biased estimation. I would be grateful if the editor could render a fair judgment on this matter and kindly take it into consideration.

Comments: I appreciate your work on climatic predictions, especially the effect of temperature on buckwheat yield, and you have examined the solar radiation after flowering predicted to have a positive impact on yield. You have shown increased temperature with a positive impact on yield is questionable as rising temperature may cause terminal heat stress, etc. So clarification is needed on how increased temperature would certainly have yield-related benefits while not having stress-related defects on crop productivity? Hence, please add some points concerning the temperature-induced heat stress-related consequences.

Response: We sincerely appreciate your evaluation of our paper. Our paper argues that increased temperatures indirectly enhance buckwheat yield through the shortening of the pre-flowering period. In fact, as you have pointed out, our study shows that an increase in post-flowering temperature directly has a slightly negative impact on yield after flowering (S1 Fig b). However, our results indicate that rising temperatures lead to a shortened pre-flowering period, which in turn results in increased day length after flowering, thereby extending the post-flowering period, increasing the cumulative solar radiation after flowering, and ultimately enhancing yield. The fact that this yield increase outweighs the negative effects of higher post-flowering temperatures is the key factor behind the estimated yield enhancement due to increasing temperatures. As this point may have been insufficiently explained in the original manuscript, we have revised the discussion and clarified the abstract accordingly.

In the discussion section, while describing the relationship between pre-flowering temperature and yield, we noticed that we mistakenly referred to an incorrect temperature range (LL. 321-322 and LL. 333 in the " Revised Manuscript with Track Changes"). We sincerely apologize for this error and have corrected it with the appropriate values.

Comments: Was the model you have used linear or a linear mixed model? What kind of dependent and predictor variables used for the model?

Response: The model used in this study is the Generalized Additive Model (GAM). In a GAM the relationship between explanatory variables and the dependent variable is expressed as a nonlinear function. This is explained in the Statistical Analysis section. However, for the reader's understanding, a brief explanation of the GAM has been added.

Comments: Have you assigned the fixed effects of predictor variables?

Response: All explanatory variables were assigned as fixed effects. This has been added to the Statistical Analysis section.

Comments: Please mention the average buckwheat yield per acre in 1994, 2001, and for any recent year, as you have mentioned the number of sample sizes only and not the yield.

Response: The overall average, the average for 1994 (the year with the smallest sample size), and the sample size for 2001 (the year with the largest sample size) have been added to the Past crop yield and calendar data section.

Comments: Why was the coefficient of determination (R²) weakly positive with a mere 0.23 of observed and estimated yields?

Indeed, while the estimated R² is low, this is attributable to the deliberate adoption of a simple model in our statistical analysis. In our study, we utilized nationwide data from across Japan to examine the relationship between meteorological factors and yield, estimating a single spline function to represent this relationship uniformly throughout the country. This approach was intentionally chosen to address extrapolation issues in future predictions.

Although one might reasonably expect that modifying the model—for example, by estimating distinct spline functions for each municipality, as is common in many other studies—would result in a higher R², such modifications would introduce extrapolation problems when future temperature increases extend beyond the range of the training data used to construct the statistical model.

Because our model is purposefully kept simple, it inherently possesses a low degree of freedom (i.e., limited flexibility), which in turn leads to a fundamentally lower R². Nonetheless, if a statistically significant relationship between meteorological factors and yield can be detected even with such a simple model, we can confidently assert the robustness of this relationship.

Since this point is important, we have added this discussion to the final paragraph of the "The relationship between climatic factors and yield" section.

Comments: There are typos, for instance in line 57, the recent IPCC report [3]...

Response: Thank you for your comment. We have made the corrections.

Comments: Please increase the resolution of the figures.

Response: The resolution of the figures uploaded to PlosOne is high; however, it appears that the resolution is reduced when PlosOne combines the figures with the main text to create the PDF. If the resolution of the uploaded figures itself is actually low, please let me know.

Response to the reviewers.

Response to Reviewer #2:

We sincerely appreciate the reviewers' valuable comments and suggestions, which have greatly helped us improve the quality of our manuscript. Their insightful feedback has guided us in refining our analysis and presentation.

Comments: This study analyzed how buckwheat yields are affected by temperatures and day length based on long-term data from all over Japan. The results indicated that rising temperatures speed up development and increase day length in the flowering period, thus increasing yield as a result. However, because the analysis is too simple, there are various problems and the results are unreliable.

The authors state the advantages of a simple model, but it is an oversimplification to state that the analysis without confounding factors would still produce qualitatively reliable results.

For example, buckwheat varieties are geographically unevenly distributed, which may create a spurious correlation with temperature. Also, any temporal trends in fertilizer application would affect yields, which might also show a spurious correlation with rising temperature in these decades. Furthermore, this study only deals with phenology and yield as crop data, but buckwheat plant size (biomass) is highly important. It is possible that higher temperatures could have increased plant size in a short term, resulting in higher yields. This surely have nothing to do with the effect of daylength after flowering.

If you really want to know the effects of temperature and day-length on buckwheat growth and yield, experiments under controlled environments will be able to uncover the cause-and-effect relationship, not by analyzing data that is full of uncertainties.

Answer: While it is true that our paper employs a simple statistical model, the primary objective of our study is to model the relationship between meteorological factors and both the growth rate and yield of buckwheat using historical crop data, and to discuss how buckwheat’s growth rate and yield might be affected under future climate scenarios. If one were to criticize our work solely on the basis of the model’s simplicity, then by the same token, every statistical study based on historical data would be rendered valueless. Moreover, experimental studies under controlled conditions are not without their own limitations. In practice, differences in fertilizer application rates, the varieties used, as well as variations in planting dates, daylength, solar radiation, and temperature, make it difficult to investigate every possible combination experimentally; indeed, very few such comprehensive studies exist. We believe that a scientifically rigorous approach should not exclude one method in favor of another; rather, combining the results of controlled experiments with statistical analyses of real-world crop data brings us closer to reliable conclusions. In fact, statistical analyses of crop yield data have already produced many valuable results in the field of climate change research, and several studies have shown that the predictive power of simple statistical models is comparable to that of process-based crop models.

We appreciate the reviewer’s constructive criticism regarding the soundness of our statistical analysis from a methodological perspective. Specifically, we agree that careful consideration must be given to potential biased confounding factors, such as the geographical heterogeneity of buckwheat varieties and the temporal trends in fertilizer application, as pointed out by the reviewer.

First, although the geographical heterogeneity of varieties is not explicitly included in our model, it is implicitly incorporated. Because we employ spline functions to model the relationship between temperature and crop yield, the spline curve corresponding to each temperature range essentially represents the functional relationship between temperature and yield for the regions encompassed by that range. For example, if buckwheat exhibits low temperature sensitivity at lower latitudes and high sensitivity at higher latitudes, the spline curve for the higher temperature range would tend to be nearly flat, whereas the spline for the lower temperature range would display a pronounced relationship with temperature. Admittedly, the model does not explicitly account for fine-scale heterogeneity among varieties; however, if such heterogeneity were large—leading to markedly different responses among varieties—the estimated spline curves would tend to be parallel (i.e., indicating little response to temperature or solar radiation). Conversely, if the varieties, despite some heterogeneity, respond in roughly the same manner, the estimated spline curves would not be parallel but would instead exhibit clear slopes with respect to temperature and solar radiation. In any case, by using a simple statistical model as we have, any unbiased confounding factors will simply result in larger error terms rather than artificially imposing a relationship between meteorological factors and yield. In that sense, our analysis remains robust.

Furthermore, because temporal trends in fertilizer application could serve as a biased confounding factor, we have incorporated a spline function of year into our model. Although the amount of fertilizer applied may change from year to year, any factors that change gradually over time, independent of meteorological conditions, are absorbed by this “year” term. Consequently, any anthropogenic factors that gradually change over the years are accounted for by this component, regardless of the absolute fertilizer amounts.

It is true that statistical data inherently carry uncertainty; however, statistical methods are precisely the tools we use to extract the meteorologically relevant signals from within this random variability.

Response to Reviewer #3:

Comments: I would like to thank the authors for their effort in conducting this valuable study on buckwheat, a crucial crop with significant nutritional and agricultural benefits. Buckwheat plays a vital role in food security, especially under climate change conditions, and understanding its phenological responses to environmental factors is essential for optimizing yield. This research provides meaningful insights into how climate variability influences buckwheat productivity, which is critical for future agricultural planning and adaptation strategies.

Answer: We sincerely appreciate the reviewer for taking the time to review our manuscript. Your thoughtful feedback has been invaluable in enhancing the quality of our work. We are honored by your evaluation and grateful for your insightful comment.

---

## [Decision Letter · Decision Letter 2]

19 Feb 2025

PONE-D-24-13382R2Complex relationship between crop yields and crop growing period: The shortened growing period before flowering contributes to yield increase in buckwheatPLOS ONE

Dear Dr. Sakurai,

Thank you for submitting your manuscript to PLOS ONE. After careful consideration, we feel that it has merit but does not fully meet PLOS ONE’s publication criteria as it currently stands. Therefore, we invite you to submit a revised version of the manuscript that addresses the points raised during the review process.

We look forward to receiving your revised manuscript.

Kind regards,

Karthikeyan Thiyagarajan, PhD

Academic Editor

PLOS ONE

Journal Requirements:

Additional Editor Comments:

Dear Authors,

I appreciate your appropriate responses and revisions. However, I have a few suggestions to improve your manuscript slightly better than the latest version.

Please mention the scientific name for Buck wheat (Is it a common Buck wheat or tartary Buckwheat?) in title and at least in introduction and in required lines.

If you wish, you could check some articles for the photosynthesis efficiency and quantum yield with C3 and C4 plants:

https://ps.ueb.cas.cz/pdfs/phs/2020/01/06.pdf

https://pmc.ncbi.nlm.nih.gov/articles/PMC1066505/

I found a reference indicating the positive correlation (though not significant) between the yield and temperature in buckwheat; please check it: https://www.mdpi.com/2073-4395/11/12/2371

Reviewers' comments:

Reviewer's Responses to Questions

**Comments to the Author**

1. If the authors have adequately addressed your comments raised in a previous round of review and you feel that this manuscript is now acceptable for publication, you may indicate that here to bypass the “Comments to the Author” section, enter your conflict of interest statement in the “Confidential to Editor” section, and submit your "Accept" recommendation.

Reviewer #3: All comments have been addressed

2. Is the manuscript technically sound, and do the data support the conclusions?

Reviewer #3: Yes

3. Has the statistical analysis been performed appropriately and rigorously? 

Reviewer #3: Yes

4. Have the authors made all data underlying the findings in their manuscript fully available?

Reviewer #3: Yes

5. Is the manuscript presented in an intelligible fashion and written in standard English?

Reviewer #3: Yes

6. Review Comments to the Author

Reviewer #3: Thanks for your work on buckwheat; I would like to hear more about your advancement in promoting the buckwheat agronomy.

7. PLOS authors have the option to publish the peer review history of their article (what does this mean?). If published, this will include your full peer review and any attached files.

Reviewer #3: **Yes: **Hassona, M.M.

---

## [Author Response · Author response to Decision Letter 3]

19 Mar 2025

Response to the editor

We sincerely appreciate the detailed and insightful comments provided by the editor regarding our manuscript revisions. We are deeply grateful for your thorough examination of our work and for introducing the references that have enriched our discussion. With the kind guidance of the editor, our paper has been greatly improved and refined.

Comment: Please mention the scientific name for Buck wheat (Is it a common Buck wheat or tartary

Buckwheat?) in title and at least in introduction and in required lines.

Answer: Thank you for your important comments. I have inserted the scientific name and species name into the title and added it in the main text where it was deemed necessary.

Comments: If you wish, you could check some articles for the photosynthesis efficiency and quantum yield with C3 and C4 plants:

https://ps.ueb.cas.cz/pdfs/phs/2020/01/06.pdf

https://pmc.ncbi.nlm.nih.gov/articles/PMC1066505/

I found a reference indicating the positive correlation (though not significant) between the yield and temperature in buckwheat; please check it: https://www.mdpi.com/2073-

4395/11/12/2371

Response: Thank you very much for providing the valuable information on the papers, which was extremely helpful in revising the discussion section of the manuscript. These papers played a crucial role in addressing the consistency of our results. I have updated the discussion section, citing the papers you introduced (LL. 318-344 and LL. 399-403 in Revised Manuscript with Track Changes). In addition, we found an error in the paragraph of LL.318 (in Revised Manuscript with Track Changes) where "after flowering" was mistakenly written instead of "before flowering," and we have corrected it. We apologize for this oversight.

---

## [Editor Report · Decision Letter 3]

24 Mar 2025

Complex relationship between crop yields and crop growing period: The shortened growing period before flowering contributes to yield increase in common buckwheat (Fagopyrum esculentum)

PONE-D-24-13382R3

Dear Dr. Sakurai,

We’re pleased to inform you that your manuscript has been judged scientifically suitable for publication and will be formally accepted for publication once it meets all outstanding technical requirements.

Kind regards,

Karthikeyan Thiyagarajan, PhD

Academic Editor

PLOS ONE

Additional Editor Comments:

Dear Authors,

After careful scientific evaluations with peer reviews, I am pleased to confirm the manuscript entitled "Complex relationship between crop yields and crop growing period: The shortened growing period before flowering contributes to yield increase in common buckwheat (Fagopyrum esculentum)." has been accepted for publication in PLOS ONE.

Kind regards,

Karthikeyan Thiyagarajan PhD

Academic Editor, PLOS ONE.

---

## [Editor Report · Acceptance letter]

PONE-D-24-13382R3

PLOS ONE

Dear Dr. Sakurai,

I'm pleased to inform you that your manuscript has been deemed suitable for publication in PLOS ONE. Congratulations! Your manuscript is now being handed over to our production team.

Kind regards,

on behalf of

Dr. Karthikeyan Thiyagarajan

Academic Editor

PLOS ONE